# Reviewing and Integrating AEC Practices into Industry 6.0: Strategies for Smart and Sustainable Future-Built Environments

**Amjad Almusaed** [1,*], **Ibrahim Yitmen** [1] **and Asaad Almssad** [2]

1   Department of Construction Engineering and Lighting Science, School of Engineering, Jönköping University,
    551 11 Jönköping, Sweden; ibrahim.yitmen@ju.se
2   Department of Engineering and Chemical Sciences, Karlstad University, 651 88 Karlstad, Sweden;
    asaad.almssad@kau.se
*   Correspondence: amjad.al-musaed@ju.se; Tel.: +46-700451114

**Abstract:** This article explores the possible ramifications of incorporating ideas from AEC Industry 6.0 into the design and construction of intelligent, environmentally friendly, and long-lasting structures. This statement highlights the need to shift away from the current methods seen in the AEC Industry 5.0 to effectively respond to the increasing requirement for creative and environmentally sustainable infrastructures. Modern building techniques have been made more efficient and long-lasting because of AEC Industry 6.0's cutting-edge equipment, cutting-edge digitalization, and ecologically concerned methods. The academic community has thoroughly dissected the many benefits of AEC Industry 5.0. Examples are increased stakeholder involvement, automation, robotics for optimization, decision structures based on data, and careful resource management. However, the difficulties of implementing AEC Industry 6.0 principles are laid bare in this research. It calls for skilled experts who are current on the latest technologies, coordinate the technical expertise of many stakeholders, orchestrate interoperable standards, and strengthen cybersecurity procedures. This study evaluates how well the principles of Industry 6.0 can create smart, long-lasting, and ecologically sound structures. The goal is to specify how these ideas may revolutionize the building industry. In addition, this research provides an in-depth analysis of how the AEC industry might best adopt AEC Industry 6.0, underscoring the sector-wide significance of this paradigm change. This study thoroughly analyzes AEC Industry 6.0 about big data analytics, the IoT, and collaborative robotics. To better understand the potential and potential pitfalls of incorporating AEC Industry 6.0 principles into the construction of buildings, this study examines the interaction between organizational dynamics, human actors, and robotic systems.

**Keywords:** Architecture, Engineering, and Construction (AEC) Industry 6.0; sustainable smart buildings; human-centric design; additive manufacturing; built environments

## 1. An Introduction to Thematic Area Interaction: The AEC Framework in the Context of Industry 6.0

The Architecture, Engineering, and Construction (AEC) Industry 6.0 is the sixth transformative phase of the architectural, engineering, and construction sector, built on Industry 5.0 [1]. Industry 6.0 pioneers advancements in quantum computing, nanotechnology, artificial intelligence, and cloud-based energy solutions. Harmonization facilitates design, building, and maintenance processes, improving efficiency, accuracy, and sustainability. Construction has changed, like the 5.0 framework. Industry 5.0 uses AI and robotics to boost productivity, creativity, and supply chain alignment. Even though Industry 5.0 is still developing, this alignment created Supply Chain 5.0 [2,3]. This hybrid approach allows architects and engineers to pioneer sustainable structures and shape future environments. The 5.0 project management strategy uses cutting-edge tools and technology for real-time monitoring and data-driven decision-making. Smart sensors at building sites can track

progress, resource use, and site safety [4]. AI-driven algorithms can identify operational bottlenecks, optimize processes, and improve project effectiveness [5]. Analytical insights enable Project managers to prevent problems and meet deadlines [6]. Industry 6.0 is on the horizon as Industry 5.0 approaches. Industry 6.0 principles influence building design and management. A vast network of interconnected devices and sensors is needed to support innovative structures [7], allowing them to autonomously manage and improve their functions. AI, robotics, and IoT will shape Industry 6.0. These technologies will improve intelligent infrastructures. Augmented intelligence (AuI) was born when AI and HI merged before Industry 5.0 [8,9]. AI, robots, and IoT in smart buildings define Industry 6.0. Intelligent, green, and efficient infrastructures improve comfort, efficiency, and productivity. Construction 5.0, Operator 5.0, Society 5.0, human-centricity, sustainability, and resilience are incorporated into Industry 6.0. Industry 6.0 introduces interconnected and intelligently created settings by combining human abilities with technical marvels [10]. Human labor and AI-infused technology in Industry 5.0's intelligent building context improve design, construction, and management.

AEC Industry 6.0 emphasizes productivity using AI. AI systems produce design options from massive databases of environmental variables, human needs, and building performance indicators. These options can meet energy efficiency, structural soundness, and aesthetic standards [11,12]. The generative design allows infinite possibilities in Industry 6.0. AI algorithms can produce several design iterations given parameters. These algorithms can try different combinations to find novel solutions. Deep learning improves intrusion detection systems (IDSs) [13]. Architects may refine AI-generated concepts using their experience and aesthetics to create unique and refined designs. Advanced modeling and analytical tools simulate and assess lighting, sound, and airflow in Construction 5.0 [14]. Before building, architects must consider comfort and structure. These models help evaluate energy use, environmental impact, and building longevity. AI systems recommend green improvements based on energy use and building material attributes. These insights can help architects reduce energy, carbon, and waste. Industry 6.0 promotes sustainable design with renewable energy, energy-efficient materials, and intelligent technologies. Construction 5.0 relies on VR/AR. VR gives stakeholders a tangible sense of the project [15]. AR overlays digital information in the physical world, improving real-time visualization. These technologies improve communication, design iterations, and mistake reduction [16]. Industry 6.0 promotes cross-disciplinary collaboration. AI systems let architects collaborate worldwide. A study on AI for smart building design found that accessible information and idea exchange promotes interdisciplinary collaboration [17] (See Figure 1).

The fifth industrial revolution emphasizes eco-friendly design [18]. Sustainable design entails evaluating environmental data, energy consumption patterns, and the characteristics of materials. When integrated throughout development, these insights enable eco-friendly infrastructures powered by renewable energy, energy-efficient materials, and smart technology. AI-enabled robots aid architecture design inventiveness, simulation, and performance optimization. These robots can quickly calculate and design solutions that meet requirements. Dialog between architects and these robots improves innovation, creativity, and efficiency. Industry 5.0's definition and scope must be clarified to grasp its technological components and basic ideas [19]. Finally, AI systems analyze building performance measurements and energy usage patterns to construct "smart" structures. Intelligent buildings use innovative technologies to enhance building operations, ensuring occupant safety and comfort [20,21]. Advanced construction robots with sensors and AI algorithms help humans accomplish tedious and dangerous tasks.

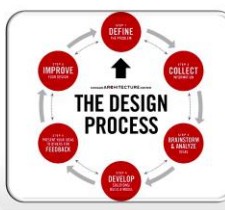

## In Design Process

1. Optimization of Designs and architectural concept
2. Generative new Design concept
3. Improving the efficiency of the design process
4. Visualization of ideas by using Virtual Reality and Augmented Reality

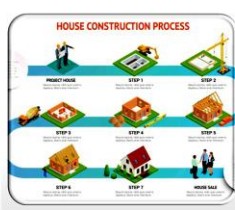

## In Construction Process

- Activation of Co-creation and cooperation concepts
- Activate the concept of sustainable building
- Enhancing Creativity, Efficiency, and Informed Decision-Making

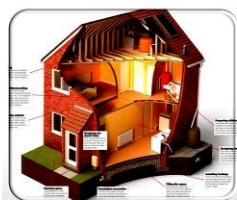

## In using Process

- Experience for Occupants
- Maintenance and Operations
- Energy Management

**Figure 1.** The AEC Industry 6.0 on the smart Building's interaction.

Architectural designs reflect humans' morals and stories. Alves et al. (2023) identify the Fifth Industrial Revolution as human-centered and sustainable [22]. Singh, Goyat, and Panwar (2023) said Industry 4.0 requires IoT, AI, and blockchain [23]. Industry 5.0 emphasizes human-centered construction. This trend prioritizes construction services and HR technology [24]. Kim (2022) advances sustainable buil§ding energy technologies [25]. AEC-FM's future depends on BIM-FM data exchange and Digital Twin technologies [26,27]. AI and IoT in construction improve stakeholder collaboration in Industry 5.0. Ikudayisi, Ayodele Emmanuel, et al. (2023) emphasize holistic building [28]. Digital (2021) notes that digital technology increases AEC visualization and sustainability [29]. This article reviews AEC Industry 6.0's disruptive effects on sustainable development. It will examine how digital technology, quantum theories, and advanced manufacturing affect AEC industry practices and professional culture. Sustainability, integrated tech uptake, and human-centric paradigms will guide the AEC 6.0 trajectory. Evaluation is conducted on quantum radar systems, cloud-integrated BIM, and the fusion of technology with human creativity. AEC's future should combine technical innovation with human ingenuity for optimal performance and sustainability.

Industry 6.0 technologies within the Architecture, Engineering, and Construction (AEC) sector have notable advantages. They provide extensive interconnectivity, optimizing worldwide partnerships. The integration of digital twins combines physical structures with real-time digital knowledge. The use of antifragile design methodologies has the potential to provide constructed environments that exhibit resilience and adaptability. The enhancement of building processes is facilitated by prioritizing software qualities such as openness and security. In general, this technological phenomenon enhances human knowledge and skills and the capabilities of construction equipment, creating opportunities for novel and environmentally conscious possibilities.

On the other hand, the rise of Architecture, Engineering, and Construction (AEC) Industry 6.0 calls for significant investment in the economic, social, and technological infrastructures to ensure smooth integration. These technological developments can alter the nature of the workforce in the AEC industry, making certain positions obsolete and

upending others. This change might exacerbate already-existing socioeconomic inequities and eliminate job possibilities, particularly for individuals who lack knowledge of this cutting-edge technology. Furthermore, the widespread use of AEC 6.0 can worsen environmental problems by causing resource depletion and more pollution. Such consequences may threaten the sustainability of artificial ecosystems and the welfare of future generations. This study offers a crucial investigation of the revolutionary effects of Industry 6.0 on the Architecture, Engineering, and Construction (AEC) industry. This research examines the progression from Industry 5.0, focusing on incorporating cutting-edge technologies such as artificial intelligence (AI), quantum computing (QC), and nanotechnology. This study argues for improved environmental flexibility, energy efficiency, and aesthetic quality via AI-driven generative design. Integrating virtual and augmented reality and cross-disciplinary cooperation highlights the emergence of "AEC 6.0" as a paradigm that prioritizes human-centricity and environmental stewardship in building efforts. This article provides a comprehensive overview of the potential technological advancements and associated challenges, including ethical dilemmas and employment implications. It is a fundamental resource for stakeholders envisioning a future where technology seamlessly integrates with human creativity to foster sustainable and resilient built environments.

## 2. Review of Industry 6.0 Perspectives and Their Implementation in the AEC Sector

### 2.1. Industry and Society 1.0–6.0 in the Context of Development

- Society 1.0–6.0 Evolutions

Technology has transformed civilization from 1.0 to 5.0. Each culture is briefly discussed. Society 1.0—the first human civilization—had rural settlements and pre-industrialization. Since the Internet's invention, technology has improved dramatically [30]. Physical labor powered the economy without technology or communication. Agriculture fueled the economy, and longstanding norms stratified and governed society. Society 2.0 began with industrialized mass production [31]. Industrialization and mechanization transformed agriculture, manufacturing, transportation, and urbanization, creating Society 2.0. Society 3.0 and Industry 3.0 and 4.0 ushered in the digital age with the widespread use of computers, the Internet, and related technology. This study investigates digital technologies that are changing society [32]. In Society 3.0, ICT has helped firms go digital, internet commerce grew, and social media exploded. Information and communication have democratized the game. Public sector organizations must provide critical services and leverage cutting-edge techniques like artificial intelligence analytics that other industries have adopted [33]. They imagine Society 5.0. Technology and economic success remedy social issues in its compassionate world. AI, IoT, and robotics will help Society 5.0 address aging populations, environmental sustainability, healthcare, transportation, and more. Technology can also improve society. Society 6.0 envisions a future where digital and real-world solutions balance economic growth and social challenges [34].

- Industry 1.0–6.0 of the Industrial Revolution

Over three centuries, new technology and industrial methods were introduced in five phases, starting in the 18th century. Steam-powered machinery began making items around 1780, according to Dixit and Uday Shanker (2023). Crafting and homesteading became common during this critical historical time [35]. Factory systems originated alongside the First Industrial Revolution. The Second Industrial Revolution introduced mass manufacturing and scientific administration to industry. CNC machinery and robotics enabled industry automation during the Third Industrial Revolution. Finally, Industry 4.0—the Fourth Industrial Revolution—incorporates computer science and information technology into production. Long-term sustainability is also emphasized. Digital manufacturing, additive manufacturing, and cyber-physical systems promise to improve connectivity and communication while simplifying data to optimize products and processes, so many developed countries have invested heavily in intelligent manufacturing technologies [36]. Due to this change, more people are working remotely.

Modern industrial technologies facilitate the reconnection to nature. An important aspect of Industry 5.0 is the seamless collaboration of people and robots. M. Ergün and his colleagues emphasize this, claiming that computer-vision approaches may effectively address human-caused mistakes and difficulties. Using computer-aided techniques may improve this even further [37,38]. Experts are now more concerned with improving the customer experience due to the development of customized and cooperative robots instead of just striking a balance between human wants and what technology can provide.

The cyclical nature of industrial revolutions and altering manufacturing paradigms show that this dynamic has evolved. Technology changed relationships [39]. Industry 6.0 is a paradigm shift based on quantum breakthroughs, advanced biotechnologies, neural integration, decentralized autonomous systems, and a strong focus on sustainable production [40]. Figure 2 shows that this current iteration emphasizes resilience, human-centeredness, sustainability, and manufacturing methods.

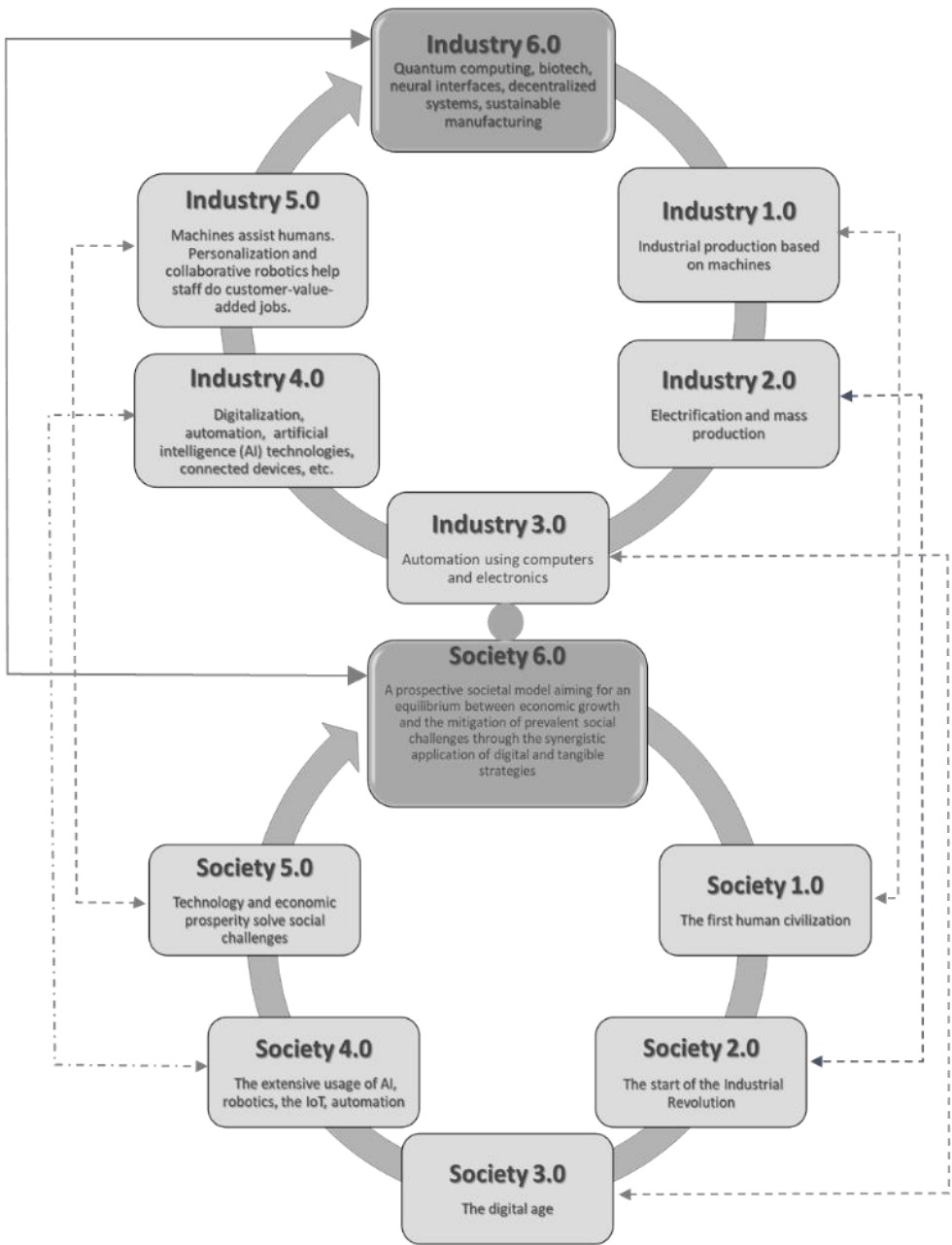

**Figure 2.** The Evolution of Interactions between Industry and Society: From Version 1.0 to 6.0.

## 2.2. Benefits of AEC Industry 6.0 in Architecture and Construction

Industry 6.0 makes design and construction more connected and promotes dynamic industry networks. It encourages flexible supply chains, adaptable value networks, and unprecedented global information interchange [41]. AEC Industry 6.0 emphasizes creating a learning and equilibrium-oriented environment. Physical and virtual architectural and infrastructure aspects inspire a new approach to integrated project delivery [42]. Stakeholders can see and evaluate physical structures and virtual data to improve global cooperation, technical assistance, and decision-making. In AEC Industry 6.0, antifragile design prioritizes adaptability and strength. Unlike conventional buildings, this forward-thinking sector encourages a resilient system design that can resist and profit from shocks. Industry 6.0 considers functional and non-functional needs for architectural and construction systems, including openness, usability, security, and mobility [43]. These barriers now aid sophisticated projects. AEC Industry 6.0 should stimulate design and construction innovation. Human abilities and cutting-edge technology will make built settings more sustainable, effective, and beautiful [44]. These revolutionary methods will elevate architectural design and construction, enhancing human and environmental well-being.

## 2.3. Additive Manufacturing (3D Printing)

Production, distribution, and use processes are dramatically changing thanks to additive manufacturing (AM) technology, particularly in the architectural, engineering, and construction (AEC) industry [45]. This ground-breaking paradigm change, including a broad range of technologies, significantly impacts the production and use of a wide range of goods and parts. According to S. Salinas Monroy, P. Li, Y. Fang, and K. A. Polarities, this revolutionary movement created consumer goods and radically altered the architectural design field and the building sector. The core of the AM paradigm, 3D printing, uses a systematic process of successive layer-by-layer material deposition. This method of operation significantly lowers marginal manufacturing costs and promotes dramatic industrial supply chain simplification [46]. One notable benefit of additive manufacturing (AM) in the architecture, engineering, and construction (AEC) industry is its capacity to facilitate localized production [47]. The proximity of manufacturing facilities to customers can significantly alter the supply chain dynamics, improving efficiency and flexibility. Furthermore, additive manufacturing (AM) technology enables production with considerably reduced marginal costs and offers a potential pathway for enhanced economic efficiency in manufacturing and building procedures (see Figure 3).

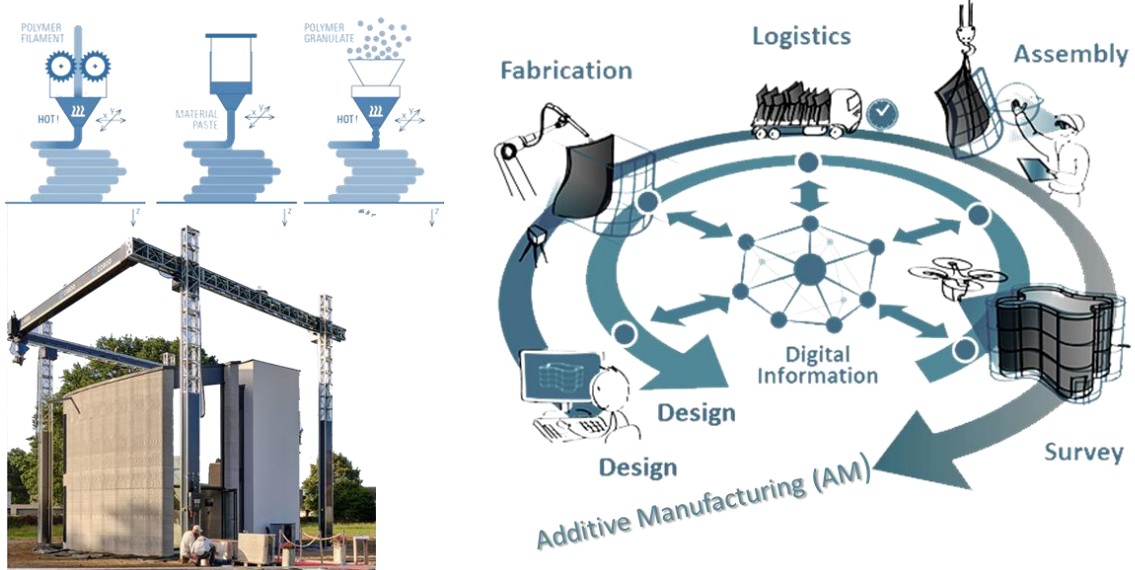

**Figure 3.** Additive Manufacturing (AM) working process in AEC Industry 6.0.

Three-dimensional printing, or additive manufacturing, is changing the AEC industry. Three-dimensional printing provides customization and accuracy for making intricate panels, fittings, and other aesthetic features [48]. This method enhances the aesthetics and utility of buildings. Many companies now 3D print entire structures. Industrial-scale printers with fast-hardening concrete save construction time, cost, and environmental impact. Apis Cor offers 24-h home printing. Urban planners also utilize 3D printing to swiftly produce and change scale models of cityscapes [49]. As 3D printing in building advances, we expect more inventive uses.

## 2.4. Artificial Intelligence and Autonomous Robots

The harmonious marriage of Artificial Intelligence and autonomous robotics fundamentally redefines AEC Industry 6.0, spearheading an organizational and operational transformation [50]. These ingenious robotic systems can tackle complex tasks, heightening efficiency, bolstering safety, and streamlining cost-effectiveness. Utilizing Machine Learning algorithms, they autonomously adapt, learn, and perfect processes. Experts Mohsen Soori, Behrooz Arezoo, and Roza Dastres point out that AI, ML, and DL have breathed new life into sophisticated robots, increasing their intelligence, efficiency, and adaptability. Advanced robotics leverage these technologies for autonomous navigation, object recognition and manipulation, natural language processing, and predictive maintenance. Additionally, they facilitate the creation of collaborative robots, or 'cobots,' that seamlessly interact with humans in dynamic environments and tasks [51] (See Figure 4).

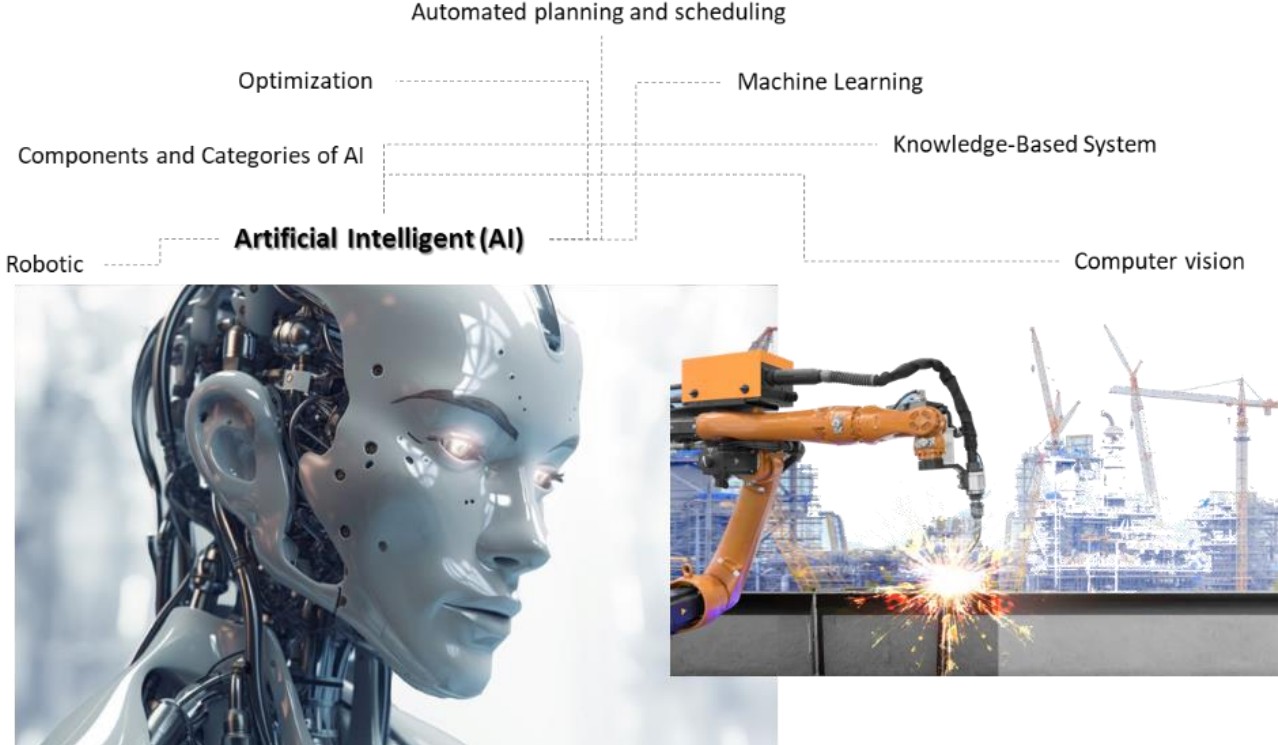

**Figure 4.** Artificial intelligence within autonomous robots' interaction.

The applications include a broad spectrum, spanning from using artificial intelligence to anticipate equipment failures in the context of predictive maintenance to the execution of precise tasks by robots in advanced manufacturing. The importance of human–robot cooperation is also underscored, with robots being used to aid human employees rather than replace them [52]. The objective of AEC Industry 6.0 is to provide a more customized, environmentally conscious, and comprehensive industrial setting by integrating human creativity with artificial intelligence capabilities [53].

### 2.5. Optimizing the AEC Industry with Cyber-Physical Systems and Smart Factory Simulations

AEC is being transformed by Cyber-Physical Systems (CPS), simulation technologies, and intelligent manufacturing facilities. The changes above affect architecture, construction, and maintenance. Specialists such as Tran Duong Nguyen and Sanjeev Adhikari assert that the construction industry requires assistance in merging the digital and tangible realms. Data interpretation and execution need to be distinct. This gap causes data fragmentation, duplication, and construction life cycle inefficiency [54]. Cyber-Physical Systems (CPS) use computer, networking, and physical processes to detect structural faults and enable predictive maintenance. Building Information Modeling (BIM) is a leading simulation technology that lets stakeholders create sophisticated three-dimensional models to simulate the complete lifecycle of a built environment. Doukari, Omar, Mohamad Kassem, and David Greenwood explain that Building Information Modelling (BIM) has evolved from a computer-assisted tri-dimensional modeling tool to include chronological scheduling, fiscal oversight, and an information management structure that can improve decision-making across the entire life cycle of constructed assets. Building Information Modeling (BIM) is a vital simulation technology tool that allows stakeholders to create sophisticated three-dimensional models and simulate a built environment's lifespan. According to Doukari, Omar, Mohamad Kassem, and David Greenwood, BIM evolved from a computer-aided tri-dimensional modeling tool to include chronological scheduling, financial oversight, and an information maze [55]. This technique optimizes performance and predicts potential problems. Smart factories within the Architecture, Engineering, and Construction (AEC) sector utilize IoT, AI, and big data analytics to enhance manufacturing [56]. Automated component production streamlines on-site assembly and reduces waste. These technologies improve architectural flexibility, safety, and efficiency. Integrating Cyber-Physical Systems (CPS) with intelligent manufacturing has transformed the construction sector, encouraging equipment and supplier collaboration. Hamzah M. et al. defined CPS as cyber-physical convergence. Interconnectedness allows global integration and substantially influences daily life, especially CPS implementation. This integration poses hurdles. CPS broadens perceptions and spurs innovation [57]. Digital twins improve building operations and help identify issues, while simulators provide a safe training environment (See Figure 5).

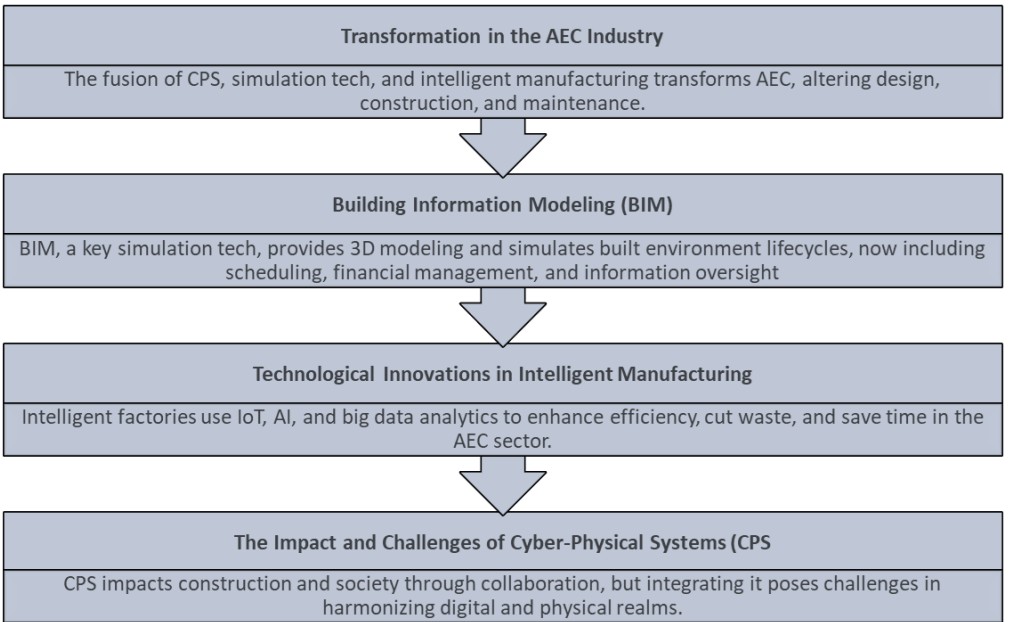

**Figure 5.** The Digital Transformation of AEC Industry 6 future: Embracing CPS, BIM, and Intelligent Manufacturing.

### 2.6. Big Data Applications in the AEC Industry 6.0 Future

In the 6.0 era, merging Big Data technology with Internet of Things (IoT) devices provides a cutting-edge technique for managing complicated and extensive building project data. Technology integration helps create novel services for architectural, engineering, and construction (AEC) professionals and others [58]. The app improves building site analysis, decision-making, and operational efficiency. Big Data can help the AEC industry understand construction patterns, building practice trends, and project abnormalities. This can spur innovation and shape design, planning, and building processes. AEC Industry 6.0 raises work roles to improve client customization and give specialists unsurpassed creativity. Automation helps architects design for clients. AEC Industry 6.0 introduced client-focused value propositions. Real-time feedback and improved automation facilitate this shift. AEC Industry 6.0 integrates human knowledge and mechanical precision [59]. Collaboration promotes decision-making, problem-solving, design, and construction. Automated work management boosts morale by offering intellectually engaging roles, personal development, and job satisfaction. Inclusion is a goal of AEC Industry 6.0 [60]. This balances career advancement, employment, and skill development. AEC Industry 6.0 encourages green building. It enables eco-friendly buildings. Adaptation will determine AEC Industry 6.0's success. Human-focused, resilient, green firms will lead. Human-oriented designs, resilient resilience, and inclusive sustainability—the pillars of AEC Industry 6.0—need more research [22].

### 3. Method and Materials

This study's approach is based on a systematic literature review, as described by RAO, Purnima; a systematic literature review aims to provide new frameworks and perspectives by critically evaluating and synthesizing existing research on a particular issue. It has been determined that an integrated systematic review is the most suitable approach to handle the research issue in this study because of its emphasis on novel and developing themes and subjects. To maintain clarity and uniformity, one must adhere to a well-defined protocol or plan during the systematic review process, setting clear criteria for study inclusion before starting the review [61]. According to CRONIN, Matthew A., a proficient integrative review holds significant potential for offering valuable insights into the present research landscape about a construction 5.0 topic while also providing recommendations for future research endeavors. The present article explores diverse categories of reviews and presents a systematic framework for conducting an integrative review.

Moreover, it offers guidance concerning the challenges encountered during the composition of integrative studies, including the equitable presentation of diverse perspectives and the synthesis of knowledge to generate novel insights. Compared to other forms of knowledge synthesis, such as narrative reviews, systematic reviews, and meta-analyses, the distinctive value of an integrative review becomes evident [62]. Dodgson (2021) emphasizes the importance of adhering to specific criteria to ensure a systematic review's methodological rigor. Moreover, the author asserts that literature reviews are crucial in advancing our understanding of a subject. Dodgson states that a literature review becomes particularly valuable when it contributes new insights to the existing body of knowledge. Although various methodologies exist for conducting literature reviews, there are shared elements of methodological rigor that should be upheld. In the present study, the SCOPUS database was employed as the primary search tool for the investigation, ensuring comprehensive coverage of the relevant literature. The search technique looked for articles published in the "construction and industry 4.0 and 5.0" category of more recently published articles that had the terms "Industry 4.0" or "Industry 4.0" in the title, abstract, or keywords. Similar procedures were carried out in the case of the words "Industry 5.0", "Society 5.0", and "Construction 5.0". While the most current keyword resulted in no relevant articles being picked (see Table 1 for an explanation of the SCOPUS search process), the top 10 pieces for each topic were chosen. In the second phase, a supplementary search was conducted using "Google Scholar" to increase the breadth and depth of the research. This study focused on

developments in the last few years and used the phrases "construction 5.0", "Industry 4.0", and "Industry 5.0", either singly or in combination. Twenty more articles were chosen in this way to make up for the need for more pedagogy-focused studies in the first search (see Table 1 for details on the Google Scholar search tactics employed). AL-ALAMI, Suhair argues that a critical mindset is necessary to study the literature. For example, when reading the literature, one must draw upon their life experiences to understand it [63].

**Table 1.** Review on the implementation of Industry 6.0 on AEC Industry.

| Analyzing Area | Subjects and Keywords | References |
|---|---|---|
| AEC (Architecture, Engineering, Construction) Industry 1–6, Smart design, and sustainability | Architectural, Engineering, Construction, Industry, foundational concepts. AI, productivity, design alternatives, interdisciplinary collaboration, intelligent building design, AI sustainable design, environmental data, renewable energy. Holistic methodologies, construction. Visualization enhancement, sustainability. Contemporary age, Industry 1, Industry 2, industrialization, Industry 3, Industry 4 Digital age, Industry 5 social challenges Industry 6.0, design and construction industry, connected environment, dynamic industry networks. | [1,7,9–12,17,18,28,29,31–36,41,41–43,50,53,56,61,62,64–87] |
| Additive Manufacturing | Adaptable value networks, The importance of adopting a critical perspective | [41,63,69] |
| The Role of 3D Printing in Building and Urban Planning | The transformative power of additive manufacturing, especially 3D printing, on the production processes in the AEC industry. The benefits of 3D printing in enhancing the design and aesthetic aspects of buildings and infrastructures | [43–47] |
| Artificial Intelligence and Autonomous Robots, Digital Twin technology, Supply Chain | Artificial intelligence (AI), robotic technologies, supply chain 5.0, operational bottlenecks, process optimization, AI algorithms. Interconnected devices, sensors, and autonomous functionalities. Blockchain. Flexible supply chains. Simplifies the industrial supply chain. Robots, collaborative robots (cobots), digital strategy. System operations, resource maximization. Algorithmic Decision Systems, personal data, correlations | [2,3,5,7,8,10,23,39,40,44,45,62,66,88–91] |
| Machine & Deep Learning and Advanced Robotics, metaverse | Artificial intelligence, human intelligence, and augmented intelligence (AuI). Robotics Deep learning, precision, and detection. Data analysis, advanced robots, productivity, safety. Decision-making, life cycle. Machine Learning, Augmented Reality, Virtual Reality, small-scale projects. Virtual environment, metaverse, augmented reality, avatars, and holograms. | [8–10,13,20,21,49,53,63,70,75,86,92–100] |
| Cyber-Physical Systems | Cyber-physical systems, connectivity, and communication. Cyber-physical procedures, Industry 5.0, Zeb, Shah, automation, | [36,55,64,91,92,101,102] |
| Building Information Modeling (BIM) as a Simulation Technology, digital twin | Smart sensors, construction sites, real-time monitoring. Real-time visualization and design iterations. Technological components, core principles. Data interchange, Building Information modeling (BIM), Facility Management (FM), physical and virtual buildings, infrastructures, and integrated project delivery. Simulators. | [4,16,19,26,27,46,55,57,88,93,94,101–104] |

**Table 1.** *Cont.*

| Analyzing Area | Subjects and Keywords | References |
|---|---|---|
| Digital Transformation | Immersive digital representations, stakeholders. Digital technologies, Industry | [16,33,45,105] |
| Big Data AI and IoT | Amalgamation, Lot, IoT, AI, Analytical capacities, Big Data, construction patterns, building practices, and project irregularities. | [10,23,34,56,58,64,90,91,99,100, 103,106,107] |
| Client-Centric Approach | Technology, long-standing customs. Client-oriented approach, real-time feedback, human expertise | [30,59] |
| Eco-friendly Construction and Sustainability | Sustainable operations<br>The importance of robots working alongside humans, aiming to create a more eco-friendly and personalized construction environment.<br>Construction sector equality, eco-friendly construction | [22,52,60,108] |
| Human-centric Design and Resilience | human-centricity, sustainable design.<br>The emphasis of Industry 5.0 is on the collaboration between humans and machines.<br>Merging human capabilities with advanced technologies for better building outcomes.<br>Human-Centric Innovation. | [10,22,24,39,44,60,86,88,109] |
| Challenges and Prospects | Project challenges, milestones, temporal, financial.<br>The challenges the building sector faces in merging digital and physical realms.<br>The challenges faced by contemporary manufacturers and the importance of maintaining equipment. | [6,54,66,75] |
| Energy Efficiency, Disassembly, circular economy, recycling. HAVC, Comfort | Energy efficiency.<br>Lighting, sound, airflow, and simulation.<br>Sustainable energy.<br>HVAC, safety, indoor air quality, energy consumption | [11,12,14,25,67,70,71,76–79,88–90,95,97,106,108,110–118] |

## 4. Transforming Construction through Industry 6.0: Technology Integration, Sustainability, and Collaborative Innovation

*4.1. Development of a Sustainable Smart Building Using AEC Industry 6.0 Principles*

The Integrated Design Process optimizes Building Information Modeling (BIM) for stakeholder engagement in modern AEC. Digital twin technologies merge virtual and physical worlds to track construction progress [88] meticulously. Dynamic supply chain methods enable real-time tracking, fast deliveries, and enhanced efficiency. Structures can better withstand natural disasters by using Antifragile Design Principles. A net-zero energy benchmark prioritizes sustainability and resource efficiency. Human-centric innovation, influenced by AEC Industry 6.0, emphasizes tenant well-being and its impact on productivity. Additionally, strict security and compliance mechanisms ensure regulatory compliance. Operator 6.0 will lead the AEC industry's subsequent development with cutting-edge technology and eco-friendly practices. This groundbreaking model uses BIM principles for real-time design collaboration, precise visual representations, and early discrepancy discovery [103]. Automation and robotics transform excavation and assembly. Real-time data interpretation via the IoT boosts operational efficiency. AI and ML provide perfect design and resource allocation. Augmented and virtual reality provide deep and engaging perspectives on varied tasks. Renewable energy shows sustainability. Cloud computing provides universal data access and advanced data analytics for informed decision-making. Figure 6 depicts a revolutionary construction and architectural design era based on this comprehensive concept.

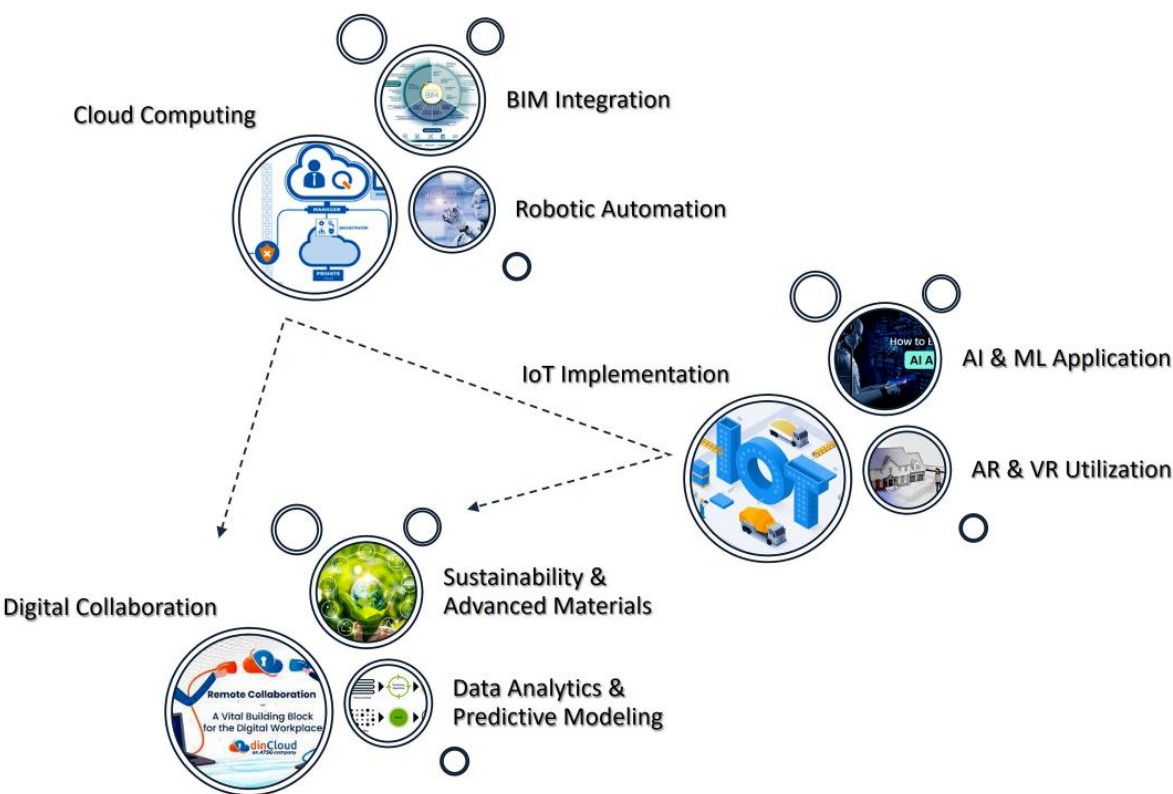

**Figure 6.** The AEC industry, Operator 6.0 for AEC Industry components.

The architectural engineering business successfully developed an advanced and environmentally friendly skyscraper by using the concepts of AEC Industry 6.0. The project exemplified the potential of technology and new approaches in revolutionizing architectural engineering, creating a building that serves as a dynamic and flexible structure, serving as a monument to the industry's future.

*4.2. Revolutionizing the Construction Sector: The Integration of Industry 6.0 and Sustainable Practices*

Architecture, infrastructure, and urban planning depend on construction. Industry 5.0 in building emphasizes human–robot collaboration [92]. AI, IoT, and blockchain affect industries, especially construction. Combining human and technology skills, Industry 6.0 boosts building productivity, sustainability, and creativity. Optimizing construction results [64], AI, IoT, VR, and machine learning may transform project planning, design, and management with Industry 5.0. These innovations improve building and growth [119]. VR revolutionizes participatory design. Stakeholder participation in design and planning improves engagement and design understanding. VR lets stakeholders plan structures realistically [93]. IoT sensors improve building maintenance, enabling data-driven design and construction. Architects, engineers, and builders must collaborate in the Fifth Industrial Revolution. AEC Industry 5.0 stresses transdisciplinary collaboration. BIM promises cost-effectiveness and high-quality design [104]. Sharing datasets and digital platforms improves project outcomes. Construction Industry 6.0 emphasizes sustainable materials and efficient procedures. Robots, AI, and 3D printing boost sustainability. Cloud computing technologies are also showing promise in modest building projects [94]. Project delivery improves with Industry 6.0. It improves operations, communication, and project engagement. Advanced technology enhances business [120] (See Figure 7).

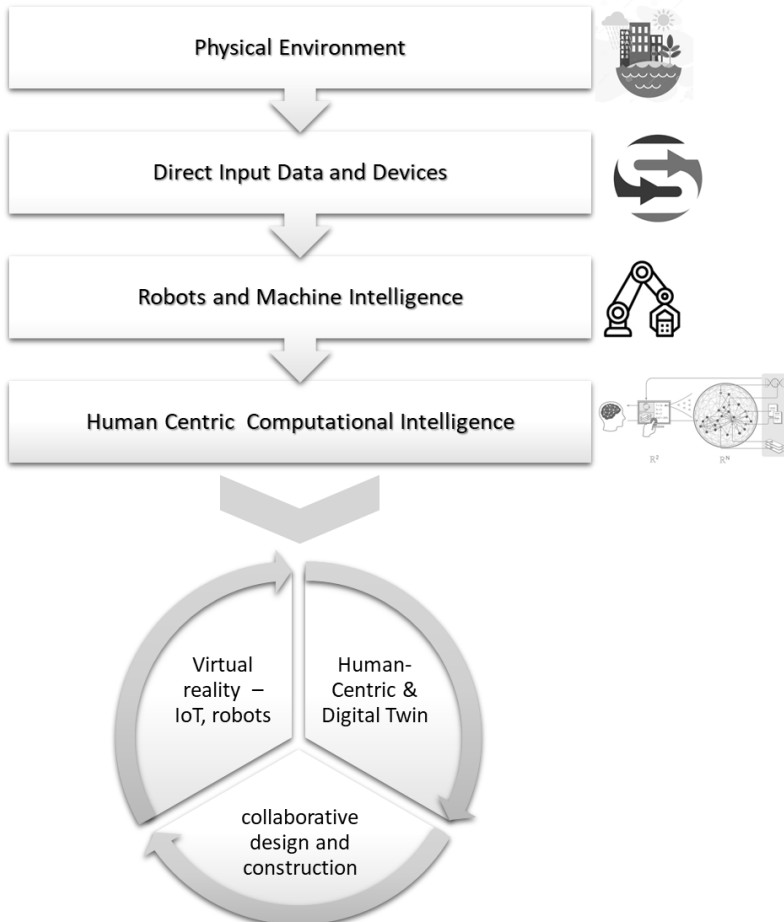

**Figure 7.** Synergizing the AEC components with Industry 6.0.

The increased usage of digital platforms and cloud-based solutions has enabled people to collaborate in real-time. Data analytics and visualization tools allow project managers to keep track of work, identify issues, and make informed choices. Dwivedi, Yogesh K., et al. (2023) affirm that a virtual environment supported by the metaverse has the potential to extend the physical world using augmented and virtual reality technologies, allowing users to seamlessly interact within actual and simulated environments using avatars and holograms [65]. Furthermore, digital twins, digital representations of tangible assets, enable simulation and predictive modeling, resulting in better project planning and risk assessment. The merger of the construction sector into Industry 5.0 has the potential to transform project conception, design, construction, and management. In the study conducted by Zairul, M. and Zaremohzzabieh, Z it is evident that waste management poses a critical challenge within the building sector due to the substantial amount of waste generated during manufacturing [66].

Moreover, the industry must address the escalating need for sustainable construction materials to accomplish the objectives outlined in the climate agreement, emphasizing the transition toward clean energy sources Utilizing cutting-edge technology, promoting cooperation, and prioritizing sustainability may help the construction sector increase production, efficiency, and innovation. However, obstacles like data security, interoperability, and workforce upskilling must be surmounted before this integration can reach its full potential. The construction industry may usher in a new age of environmentally responsible and technologically advanced building practices by adopting Industry 5.0 with the correct strategy and investments.

### 4.3. Smart building within AEC Industry 6.0

AEC Industry 6.0 emphasizes smart buildings with advanced technology and complex systems to enhance energy efficiency, occupant comfort, safety, and operations [67]. AEC Industry 6.0 is adaptable, collaborative, and flexible to create a positive human–technology relationship. Salem and Dragomir (2023) suggest adopting a risk treatment process and a unique risk matrix to improve risk management in building projects with digital twins. A digital risk management strategy improves predictive skills, helps human decision-makers reduce unforeseen expenses and failures, and boosts operational efficiency. More study into shared information and data protection is needed to minimize intentional and unintentional exploitation and create a fully digital system [121]. Smart buildings within AEC Industry 6.0 incorporate IoT devices and sensors. These advanced structures process data through analytics and machine learning. Krishnan P. et al. (2023) advocate identifying unusual energy usage to save energy. By employing this approach, smart buildings can offer a greener, sustainable setting tailored to the diverse lifestyles of residents. Optimizing energy efficiency in lights, HVAC controls, energy monitoring, building envelopes, automation systems, and renewable energy sources yields IoT device parameters. Algorithms and parameters from water, network convergence, electrical, and environmental monitoring boost energy efficiency [106]. The algorithms above enable pattern detection, trend prediction, and operational improvement. Predictive maintenance algorithms evaluate equipment sensor data to detect abnormalities and future concerns, reducing downtime. IoT and wearable technologies in intelligent buildings monitor user preferences and alter ambient variables for comfort. Sustainable renovation improves energy efficiency, building value, indoor and outdoor convenience, carbon emissions, and occupant satisfaction and well-being, according to Velykorusova (2023) [89]. Climate, lighting, and ergonomics boost productivity and satisfaction. Smart buildings monitor air quality and noise in real-time, creating healthier and more comfortable workplaces. Smart buildings have improved surveillance, access control, and emergency response systems. BIONDO, Elias Junior (2023) states that people spend 90% of their time indoors, where air pollutants like carbon monoxide (CO), carbon dioxide ($CO_2$), volatile organic compounds (VOCs), sulfur dioxide ($SO_2$), ozone ($O_3$), and nitrogen oxides (Nox) can be two to five times higher than outdoor levels and sometimes 100 times higher [110]. Video analytics, face recognition, and access controls identify security risks. Intelligent structures can initiate evacuations, guide residents to safety, and notify emergency services. Construction 5.0 smart buildings communicate with intelligent city infrastructure to share waste, electricity, and transportation data. This improves resource distribution, sustainability, and the ecological footprint. According to UTKU, Durdu Hakan et al. (2023), Digital Twin (DT) technologies generate a virtual system representation, allowing for monitoring of its operations, linkages, and interactions. Thus, fresh technical approaches and concepts are being researched to optimize building processes by maximizing restricted resources [122].

Innovative Building administration Systems (BMS) can simplify asset, facility, and energy management. Remote monitoring and control let facility managers identify maintenance issues, improve building performance, and fix problems [68]. Data analytics, current technology, and sustainability enhance operational efficiency, occupant comfort, safety, and energy saving in 5.0 smart buildings. Connectivity, automation, and sustainability define intelligent industrial infrastructure and human–machine collaboration. Smart building technologies (SBTs) offer energy efficiency, cost-effective maintenance and operation, employment opportunities, healthcare management, real-time monitoring, and improved safety and security, according to EJIDIKE, Cyril Chinonso and MEWOMO, Modupe Cecilia (2023). Understanding smart building technology's benefits is essential in developing economies. Building experts' awareness will help these regions implement and adopt these technologies [111]. Smart buildings increase comfort, energy efficiency, and operations.

I.  Automated systems: Intelligent systems and sensors optimize energy consumption in smart buildings. Real-time data, occupancy patterns, and ambient conditions optimize lighting, HVAC, and other energy-intensive equipment [95].

II. Dynamic controls: Smart buildings adapt to occupant preferences and time of day. This personalized strategy saves money by turning off lights and thermostats in empty rooms [112].

III. Data analytics and machine learning algorithms optimize power usage. These findings help facility managers manage energy and cost savings [113].

*4.4. Comfort in Intelligent Living Spaces*

Intelligent buildings allow workers to customize lighting, temperature, and other characteristics for comfort [69]. According to Majid Al Mughairi, Thomas Beach, and Yacine Rezgui, building automation systems are their brains. These advanced systems control HVAC, lighting, security, access control, surveillance, and indoor air quality. These systems collect data and execute commands via a gateway using strategically placed sensors and actuators in the building. This creates a dynamic, energy-efficient living space. These management systems balance technology and comfort by responding to occupants' unpredictable behavior [114]. This personalized approach enhances convenience, comfort, and efficacy, as thermal conditions affect thermal contentment and productivity in built environments. Bueno, A.M., de Paula Xavier, A.A., and Broday, E.E. noted that academic research has illuminated thermal comfort and productivity throughout history. Mathematical models predict productivity changes by analyzing ambient temperature variations due to the necessity to understand how environmental variables affect performance, especially considering how much time people spend indoors; multiple models have failed. Buildings must prioritize comfort, energy efficiency, and sustainability [115]. IoT technology enables real-time monitoring of air quality, noise, and other environmental factors, promoting a safer and more comfortable environment. Kureshi, R.R., Thakker, D., Mishra, B.K., and Barnes, J. assert that indoor and outdoor air quality are fundamental human rights. Yet, despite its health implications, indoor air pollution has received less focus than outdoor pollution from healthcare entities, communities, and local agencies. Methodological case studies on indoor air pollution's sources and effects are urgently needed due to the amount of time people spend indoors and its negative impacts, especially on those with respiratory and other health issues. These studies should also examine ways to modify indoor activities that cause indoor air pollution and respiratory problems [90]. IoT devices and sensors enable real-time air quality, noise, and environmental monitoring. Thus, informing building users of concerns may improve safety and comfort.

Smart Building Adaptive Systems: These buildings can automatically adjust lighting, temperature, and other systems based on occupancy. No human interaction is needed because the system is adaptive, ensuring comfort. Gholamzadehmir, Maryam, and others have popularized intelligent architecture and socially resilient cities. Building automation systems help control energy generation, use, and storage. Building automation and control systems use old and innovative control approaches. Intelligent buildings require advanced solutions [70]. Smart buildings employ building management systems (BMS) to monitor and regulate multiple building processes, improving efficiency. Lam, K.H., To, W.M., and Lee, P.K.C. found that intelligent buildings can minimize energy use and provide a responsive, pleasant, and productive interior environment. Smart buildings require an intelligent building management system (SBMS). An SBMS must execute many tasks and give the expected benefits [96].

Automated lighting controls and resource allocation in intelligent buildings improve process efficiency. Shah, S.F.A. et al. showed that computerized methods could expedite labor processes, reduce human intervention, and boost productivity [91,107]. Machine learning can automate many processes using intelligent, developing algorithms. Smart Buildings, an IoT application, is notable. These structures provide an efficient and convenient ecology through integrated operations. Baduge, Shanaka Kristombu, and their colleagues recently expanded this notion. They examine how the construction sector is adopting Digital Twin (DT), Building Information Modelling (BIM), AI, IoTs, and Smart Vision (SV). These current building and construction technologies combine efficiency,

productivity, accuracy, and safety [101]. Advanced production systems, cyber-physical methodologies, and digital technology alter the building and infrastructure; similar to the Barnesre lifecycle from design to maintenance in AEC Industry 4.0. Key elements: According to Majid Al Mughairi, Thomas Beach, and Yacine Rezgui, building automation systems are their brains. Key elements include:

- Production Systems: 3D printing, prefabrication, and offsite manufacturing.
- Cyber-Physical Tools: IoT, robotics, and actuators.
- Computing Technologies: BIM, AI, ML, cloud computing, data analytics, Blockchain, AR, and digital twins.

Such integration fosters enhanced industrial capabilities, streamlining and intelligently managing facets like temperature, safety, and maintenance via mobile devices and computers. With the rise of IoT, intelligent buildings have become central to system integrations. Advanced HVAC controls reduce peak energy demand, optimizing energy and promoting sustainable practices [91].

Smart buildings represent a significant advancement in sustainable and efficient industrial infrastructure, as they play a crucial role in enhancing energy efficiency, occupant well-being, and operational processes. By integrating cutting-edge technologies and intelligent systems, these buildings are designed to minimize energy consumption while optimizing various aspects of occupant comfort and productivity. Incorporating innovative technologies in building systems enables precise control and monitoring of energy usage, reducing energy consumption and lowering greenhouse gas emissions. By leveraging advanced sensors, data analytics, and automation, smart buildings can adapt their operations based on real-time conditions and occupant behavior, thus maximizing energy efficiency and minimizing waste. Moreover, the emphasis on occupant well-being is a cornerstone of smart building design. These buildings are equipped with features that enhance indoor air quality, regulate lighting levels, and maintain comfortable temperatures, all of which contribute to a healthier and more conducive environment for occupants. Studies have shown that such improvements in indoor environmental quality can lead to increased productivity, reduced absenteeism, and enhanced occupants' overall well-being [71]. Additionally, smart buildings streamline various operational procedures by seamlessly integrating different systems, enabling centralized monitoring and control. This integration allows for data-driven decision-making, predictive maintenance, and optimized resource allocation. As a result, processes within the building are more efficient, responsive, and cost-effective, leading to a more sustainable and productive overall industrial infrastructure [72].

### 4.5. Sustainable Development in the AEC Industry Area

In recent years, sustainable development has grown significantly in the AEC industry [73]. A balanced approach is essential to meet current needs without jeopardizing future generations, encompassing economic growth, environmental protection, and social welfare. In the Architecture, Engineering, and Construction (AEC) sectors, "sustainable development" means employing eco-friendly techniques, promoting energy efficiency, minimizing waste, and optimizing asset performance and longevity. Schutzenhofer et al. note that the AEC sectors account for 40–60% of global raw material extraction. For continuous circular economic model implementation, it is vital to consider material, emissions, and energy. This study delves into disassembly, recovery, and recycling processes, evaluating technical efforts and costs against potential eco-impact [116]. As a result of the substantial environmental effect of building operations, the AEC sector has realized the urgent need to implement sustainable practices. The world's buildings and infrastructure significantly impact energy, greenhouse gas emissions, and resource depletion. This has instigated an impetus to integrate eco-friendly building methodologies from a project's inception to its ultimate decommissioning. Various frameworks exist to facilitate sustainable development within the AEC domain. Strategies encompass utilizing renewable energy resources, implementing efficient water management systems, and meticulously considering the building's orientation concerning natural illumination and ventilation, exemplified by

green building standards [97]. Sustainable development also incorporates cutting-edge building methods like prefabrication and modular construction to reduce waste further and maximize resource efficiency. Sustainable design also prioritizes the health and comfort of building occupants since it is the best way to ensure they are both happy and productive at work. Collaboration and integration among many stakeholders are essential for achieving sustainable development in the AEC business. Sustainable goals, practical techniques, and performance monitoring need collaboration between architects, engineers, contractors, developers, policymakers, and building occupants. Building Information Modeling (BIM) and other technological advances help sustainable design by allowing for more precise design simulations, better energy performance, and easier project management [102].

## 5. Progressive Insights and Forward-Thinking Paradigms in AEC Industry 6.0 Research

### 5.1. The Future of Industrial Automation: AEC 5.0 to 6.0 and Beyond

As technology evolves, robots with sensors and a user interface that can identify and respond to unstructured environments will become more accessible. Industrial robot–human interaction will improve. Automation and AI affect individuals and employment, including repetitive or dangerous jobs [74]. Advanced robots must safely and efficiently interact with humans. Industry 5.0 cyber-physical operations in the AEC industry improve automation, real-time data processing, and decision-making. Secure networked communication protocols need more research. Zeb, Shah, et al. note that Industry 5.0 in AEC, an evolution from Industry 4.0, pursues resilience, sustainability, and human-centric solutions in new applications. Human insight and trustworthy, intelligent cobots aim to achieve zero waste, zero defects, and mass-customized production [98]. Modern firms must adopt several methods and address many needs to compete globally, according to Rojek et al. (2023). These include lowering production costs, meeting consumer product quality needs, encouraging innovation, limiting environmental impact, and ensuring operational safety. Maintenance and manufacturing equipment reliability is crucial to solving these issues. Maintaining manufacturing equipment ensures firm continuity and productivity [75]. Maintenance length and scope must be determined to efficiently operate industrial equipment and plan spare parts, workforce, and cash. A decade of study has focused on applying AI to oversee maintenance chores. AI and ML in Construction 5.0 improve predictive maintenance and adaptive control. Intelligent algorithms that examine large data sets to make judgments are a promising research field [123]. Multi-Access Edge Computing (MEC) technology extends cloud computing to wireless access networks near end-users. MEC provides real-time, low-latency, and high-bandwidth access to radio network resources when integrated with 5G systems, enabling network operators to deliver new services and create a unique ecosystem and value chain. The Internet of Things (IoT) is a complex ecosystem of different, resource-limited physical things, according to Liyanage, Madhusanka, et al. IoT has applications in healthcare, agriculture, smart cities, automotive, and industry [99]. IoT applications require centralized cloud computing for data processing and storage. IoT devices have resource constraints, including low battery power, memory, and computing. IoT must also enable real-time, scalable apps with low latency and excellent QoE as needed. Thus, IoT applications must meet performance and efficiency requirements [100]. Logic Controllers Uncrewed aerial vehicles, autonomous ground vehicles, and robots may change production. Infrastructure, safety, and autonomy are studied, like Quantum Computing. Quantum computing may enable complicated problem-solving, system modeling, and industrial process optimization. Industrial quantum computing research is possible. Construction 6.0 may involve green production, renewable energy, and energy-efficient technologies. This research may be green manufacturing [124]. Self-driving automobiles, uncrewed aerial vehicles, and robots are projected to change industrial operations even further shortly. To maximize these technologies' potential, researchers will focus on critical areas.

- Advanced Autonomy and Decision-Making: This research seeks to produce advanced autonomous systems that can make real-time decisions in complex and changing operational settings. Complex algorithms and advanced machine-learning models are

needed for meaningful autonomy. Algorithms have long existed. Their rising use in decision-making systems warrants notice. Algorithmic Decision Systems (ADS) analyze personal data to find correlations and extract decision-making information [125]. In fully automated ADS, human involvement in decision-making may be absent. These choices affect people's credit, career prospects, medical care, and legal fines, highlighting the study's importance and complexity.

- Autonomous systems must prioritize safety and security. This study examines risk reduction, fail-safe measures, and building control systems cyber-security. According to Illiashenko et al., security-informed safety (SIS) or cybersecurity-informed safety (CSIS) is a developing interest in assessing autonomous system safety and reliability using an entropy-oriented technique [117].

- Autonomous Systems and Infrastructure Integration: Integrating autonomous systems into existing infrastructures and conceptualizing and developing new infrastructures to support them has become a significant area of research. This includes creating communication protocols, optimizing information transfer, and adapting the architectural design for emerging technology. Yogesh K. et al. have proposed that the metaverse could expand the physical realm using augmented and virtual reality technologies. Avatars and holograms may enable seamless interaction between real and virtual surroundings. However, the technology and infrastructure needed to create massive, immersive virtual worlds that allow avatars to span platforms are still in development. Despite this constraint, scholarly research into the metaverse's potential to revolutionize society and human interaction is rising, suggesting a promising route for future research and innovation [76].

- Accelerating AEC Materials Discovery and Design Quantum Computing: Quantum computing may solve complicated problems that regular computers cannot. Quantum algorithms and frameworks for AEC procedures are being studied. Supply chain optimization and advanced materials research use material passports. Quantum computing emphasizes materials science. Yue Liu et al. say contemporary research screens innovative materials with increased performance and models quantitative structure–activity relationships. These efforts represent a new frontier in materials research, where experimental studies and computational modeling are time-consuming, resource-intensive, and limited by experimental conditions and theoretical foundations. Accelerating materials discovery and design requires a paradigm shift, including computational research and experiments. Thus, material features can be better understood, enabling more efficient and focused research into novel materials. This integrated strategy may disclose key material behavior regulators, enabling more effective and purpose-driven materials discovery and design [77]. Quantum computing and AEC manufacturing may change materials science. Quantum algorithms can overcome computational limits and speed up materials research. An integrated framework that links computers and experimentation aims to develop a future with unique materials that can alter many sectors and expand human ingenuity.

- Renewable Energy Integration: AEC Industry 6.0 investigates renewable energy integration into industrial processes and supply networks. Examples include optimizing energy storage systems, developing new grid technologies, and improving industrial energy efficiency. Worku (Y) reports a rise in greenhouse gas reduction and electric energy dependability measures. Over the last decade, intermittent renewable energy sources (RESs), including photovoltaic (PV) and wind, have been added to the power system. However, operational and control difficulties make renewable energy integration difficult for the power system. Generation uncertainty, voltage and angular stability, power quality, reactive power assistance, and fault ride-through are blocks [78]. However, research will advance energy-efficient design and construction methods. Energy and waste reduction can lessen AEC businesses' environmental impact.

- Sustainable manufacturing will spur research on eco-friendly materials and techniques [108]. Engineers and scientists will focus on biodegradable materials, circular

economy approaches, and low-impact production [79]. Kazakova and Lee explain that the circular economy paradigm and changing consumption patterns force the Architecture, Engineering, and Construction (AEC) industry to embrace more sustainable design and operations. In this setting, customers and investors pay attention to firms' environmental, economic, and social behaviors. Over 20% of worldwide greenhouse gas emissions come from manufacturing. In a UN Environment Programme assessment, the production and consuming phases of the industrial cycle are highlighted as harmful. Toxic gases, acidic compounds, and unsustainable depletion of non-renewable resources worsen environmental stress [80] (see Figure 8).

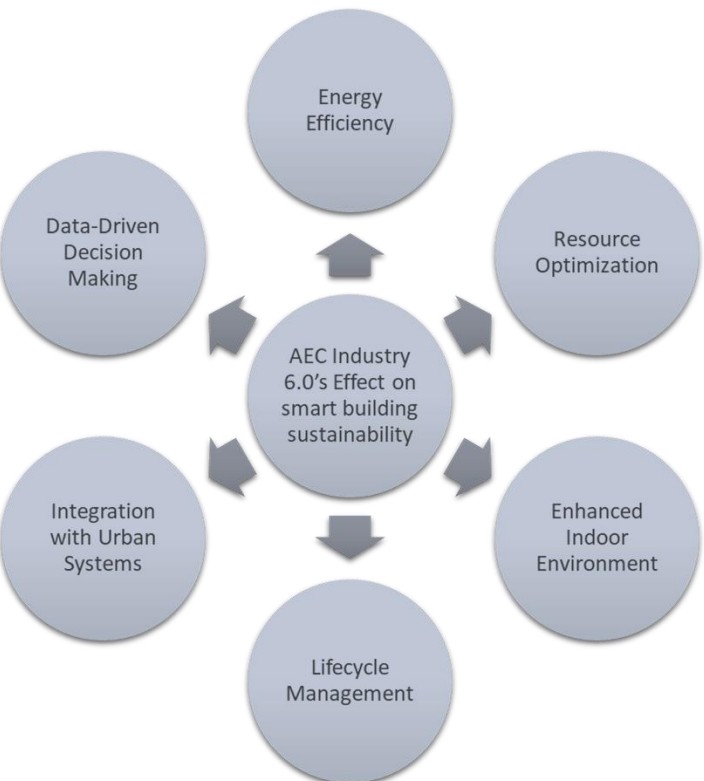

**Figure 8.** Smart Buildings Components Sustainability within the AEC Industry.

Researchers grapple with transdisciplinary challenges as technology progresses, striving to make industrial processes autonomous, efficient, and sustainable [81]. Technology's societal and ethical implications are pressing; hence, its use must ensure minimized harm and disparity as industries evolve. Crucial issues include data privacy, security, and individual rights. Technological advancements often surpass existing regulations, leading to potential pitfalls. A harmonized international legal framework, adaptive to technological changes, is imperative for safe and ethical technical integration [82]. The environmental challenges presented by technology-centric industries necessitate an urgent focus on reducing ecological footprints, promoting sustainable practices, and advancing renewable energy research. Simultaneously, data-driven businesses highlight the need for robust encryption, secure data exchange, and technologies that prioritize user privacy. Striking an equilibrium between automation and human collaboration, bolstered by intuitive user interfaces and ongoing workforce education, is pivotal for progressing societal paradigms [83]. Multidisciplinary research holds the key to addressing these multifaceted challenges. Collaboration across disciplines fosters innovation and the development of green technologies, promoting sustainable practices and the circular economy and reducing environmental impacts [84]. Technologies such as automation, AI, and data analytics optimize processes, conserving resources while boosting productivity. Tackling global challenges demands international interdisciplinary cooperation in sharing knowledge and resources. Multidisciplinary re-

search stands poised to harness technology and learning, addressing concerns like ethics, legal frameworks, sustainability, data privacy, and human–machine collaboration with a holistic approach. Embracing these opportunities and challenges can usher in an era of innovation and heightened environmental consciousness (See Figure 9).

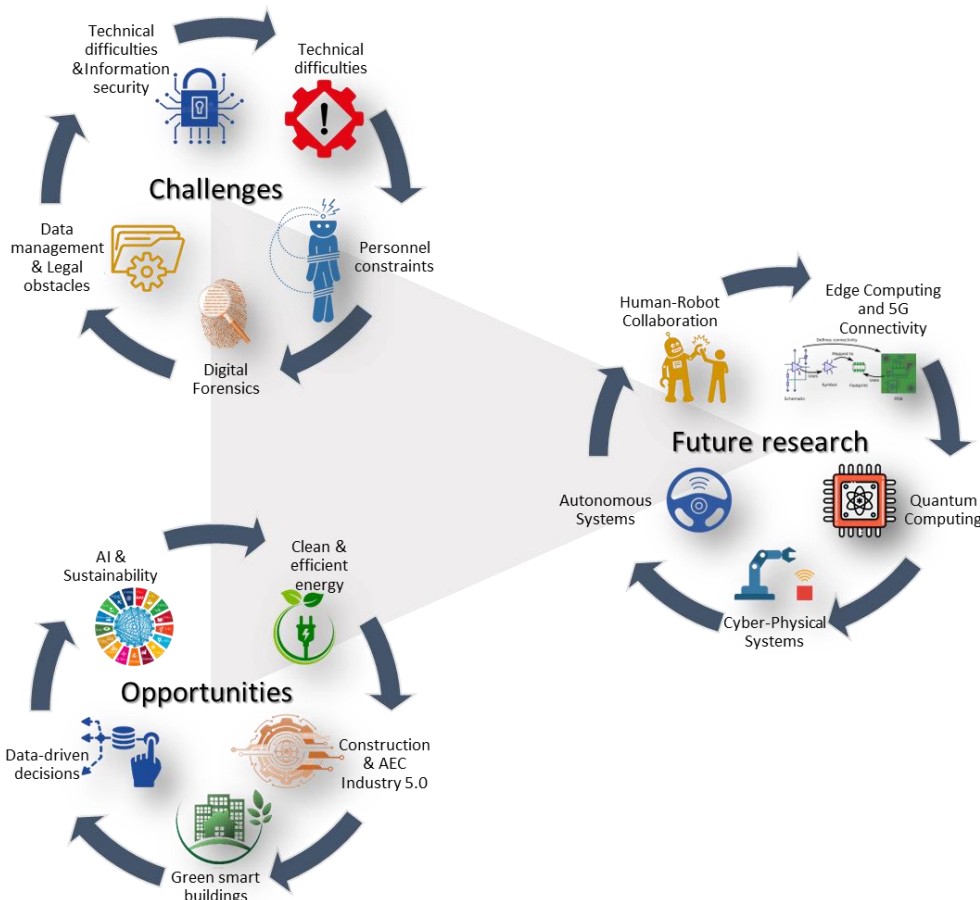

**Figure 9.** Synthesizing Challenges, Opportunities, and Future Research.

### 5.2. Human-Centric Evolution: Navigating the Transition to AEC Industry 6.0

The recent shift in organizational function has prominently featured human-centric methodologies, marking a change in the dominant paradigm. Industry 4.0 has transformed manufacturing through automation, digitalization, and connectivity [85]. AEC Industry 5.0 places humans at the center of these technological changes, utilizing technology to promote human well-being, social inclusion, and sustainable development [109]. It envisions a society where humans and robots collaborate for a brighter future.

Grabowska, S., Saniuk, S., and Gajdzik, B. have pointed to the heightened scientific interest in industrial humanization, sustainability, and resilience due to the pervasive digitization and advancement of fourth-industrial revolution technologies [109]. Industry 4.0 focuses on cyber-physical systems within organizations and their supply chains, while AEC Industry 6.0 emphasizes people's unique attributes and talents beyond robotic capabilities. The core objective is to leverage technology to enhance human abilities, granting them more autonomy.

As highlighted by Wang, Baicun, et al., human-centricity remains pivotal to Industry 5.0, underscoring the importance of human needs ranging from health and safety to self-actualization and personal growth [86]. The Human Digital Twin (HDT) concept is fundamental in achieving this focus within intelligent manufacturing systems. HDTs provide digital representations of humans, revolutionizing human–system integration by incorporating human characteristics into system design. A hallmark of AEC Industry 6.0 is

the symbiotic relationship between humans and robots. Instead of replacing humans, the objective is to augment human capabilities, enabling them to participate in advanced decision-making, creativity, and problem-solving. This industry also champions digital access equity, aiming to equip underserved communities with modern tools and training, thus reducing biases and promoting inclusivity [105]. The potential of the metaverse, which blends physical and digital interactions, is likened to past innovations such as the internet and e-commerce, offering tools to effectively manage the transition to AEC Industry 5.0 [126]. AEC Industry 6.0 prioritizes human health and safety [118]. Utilizing technology improves working conditions, enhancing quality of life. Repetitive or labor-intensive tasks can now be relegated to machines and automation, allowing employees to focus on more fulfilling roles [127]. The shift towards AEC Industry 6.0 represents a crucial turn in industrial evolution, merging human expertise with technological proficiency. This blend results in manufacturing solutions that are resource-efficient and valued by consumers, heralding a transformative shift in the industry [87].

## 6. Discussion

In Industry 6.0, the Architecture, Engineering, and Construction (AEC) sector undergoes significant changes, especially regarding intelligent buildings. This shift integrates human intelligence (HI), augmented intelligence (AuI), artificial intelligence (AI), and Internet of Things (IoT) technologies. This approach synergizes AI-driven automation with human creativity, evident in AI-assisted generative design and process optimization. This leads to collaborative efforts between architects, AI algorithms, and robotics. Integrating IoT and AI brings forth data-centric construction management methods, enhancing construction predictability and minimizing risks. The emphasis on sustainability and environmental responsibility is evident in AEC 6.0, with the EU's green transitions focusing on AI, renewable energy, and energy-efficient materials. This makes buildings more sustainable and aligns with global sustainability objectives. VR/AR technologies have significantly advanced design visualization, collaboration, and immersion, improving planning and reducing errors. Despite these advancements, challenges, such as ethical concerns and the potential impacts of automation on employment, persist. AEC 6.0 aspires to revolutionize intelligent construction by integrating various technologies and principles, promoting creativity, efficiency, and environmental consciousness. The complexities of these components necessitate continual exploration and adaptation in both theoretical and practical spheres. Future research should address practical applications, ethical challenges, and the establishment of standardized frameworks. The literature underscores the importance of human-centric design in transitioning from Industry 6.0 to AEC 6.0. The present research indicates a shift towards human–machine interaction, sustainability, and resilience, calling for a comprehensive strategy combining human ingenuity with technological efficiency. The study reviews SCOPUS and Google Scholar literature, tracing the progression from Society 1.0 to 6.0. Industry 6.0 represents a significant technological advancement in construction. Scholars, including Alojaiman (2023), emphasize themes showing Industry 5.0's transformative power in the building. Technologies like AI, IoT, blockchain, and VR have redefined the construction landscape [92]. Melnyk, Leonid Hryhorovych, et al. (2023) state that Construction 6.0 fosters technological tools and human collaboration to elevate construction productivity [64]. Lai et al. (2019) note the cross-disciplinary nature of Industry 5.0, with collaborative design enabling professionals across disciplines to innovate [104].

Current discourses prioritize sustainability. Industry 6.0 can significantly enhance sustainability in construction by employing AI, robots, and 3D printing, reducing waste and ecological impact. The literature, including works by Waqar A et al. (2023), underscores the shift toward sustainable construction techniques. Challenges, including data security and workforce upskilling, need addressing. Dwivedi, Yogesh K., et al. (2023) highlight emerging technologies and their potential impacts. AEC 5.0 and 6.0 demonstrate significant advancements in the ARC sector, with increased automation, human–robot collaboration,

and technological enhancements. As authors like Taesi, C., Aggogeri, F., and Zeb, Shah, et al. mention that industry research emphasizes improving robotic systems and integrating newer technologies. Incorporating advanced technologies in the AEC sector signifies a monumental move towards sustainable, efficient, and user-centric spaces [74,98]. This evolution requires a collective effort to maximize benefits while addressing inherent challenges.

## 7. Conclusions

The future AEC Industry 6.0 represents a significant shift from conventional construction methods, emphasizing human-centered principles over machinery. This research explored the social benefits of AEC Industry 6.0, which seeks to enhance collaboration between humans and machines through technology. It fosters inclusivity, promotes well-being, and ensures sustainability. Key components include artificial intelligence (AI), the Internet of Things (IoT), robotics, human intelligence (HI), and traditional construction methods. This synergy has birthed innovative construction techniques, improved efficiency, and increased environmental awareness. AEC Industry 6.0 integrates smart buildings, digital twins, and AI algorithms to make construction more predictive and efficient while meeting global sustainability goals and reducing environmental impacts. Challenges include skill development and ethical concerns, necessitating ample resources for education and training and a strong emphasis on stakeholder trust.

Moreover, Construction 6.0 focuses on customization, allowing collaborative robots greater autonomy, revolutionizing manufacturing, and opening new opportunities for those with specialized skills. AEC Industry 6.0 heralds an era of innovative construction, environmental dedication, and human-centric design. Through technology-driven collaboration and innovation, it holds promise for sustainability, efficiency, and positive societal impacts. Achieving these goals requires a focus on human-centered principles and addressing inherent challenges. More research is needed to grasp its effects, set guidelines, and plan ways to utilize AEC Industry 6.0's potential for creating eco-friendly and adaptable construction methods. Industry 6.0 could bring transformative changes to industries. While it promises virtualized, antifragile manufacturing and services emphasizing customer-centric strategies, dynamic supply chains, and automation-driven flexibility, job displacement due to increased automation is the primary concern. Future industrial revolutions should prioritize job creation to avoid socioeconomic discontent.

**Author Contributions:** Conceptualization, A.A. (Amjad Almusaed) and A.A. (Asaad Almssad); methodology A.A. (Amjad Almusaed) and A.A. (Asaad Almssad); software, A.A. (Amjad Almusaed); validation, A.A. (Amjad Almusaed), A.A. (Asaad Almssad), and I.Y.; formal analysis, A.A. (Amjad Almusaed); investigation, A.A. (Amjad Almusaed) and A.A. (Asaad Almssad); resources, A.A. (Asaad Almssad); data curation, I.Y.; writing—original draft preparation, A.A. (Amjad Almusaed); writing—review and editing, I.Y.; visualization, A.A. (Amjad Almusaed); supervision, A.A. (Amjad Almusaed); project administration, A.A. (Amjad Almusaed). All authors have read and agreed to the published version of the manuscript.

**Funding:** This research received no external funding.

**Institutional Review Board Statement:** This statement is excluded.

**Informed Consent Statement:** The statement is excluded, where we did not involve humans.

**Data Availability Statement:** Not applicable.

**Conflicts of Interest:** The authors declare no conflict of interest.

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
