# Peer review of "Reviewing and Integrating AEC Practices into Industry 6.0: Strategies for Smart and Sustainable Future-Built Environments"

_sustainability, doi:10.3390/su151813464_

Round 1
Reviewer 1 Report
Manuscript examines Industry 6.0 and the construction industry and the creation of structures and their possible consequences. Adding the following citations to the introduction part of the study will increase the quality of the manuscript. With the inclusion of citations, the work is eligible for publication.
* Hatır, E., Korkanç, M., Schachner, A., Ince, I. (2021). The deep learning method applied to the detection and mapping of stone deterioration in open-air sanctuaries of the Hittite period in Anatolia. Journal of Cultural Heritage, 51, 37-49.
*Hatır, M. E., Ä°nce, Ä°., Korkanç, M. (2021). Intelligent detection of deterioration in cultural stone heritage. Journal of Building Engineering, 44, 102690.
Author Response
Thank you; The two-references were added [37, 38] in section 2

Reviewer 2 Report
Dear Authors,
I’ve read your well-written article. The topic is relevant and very interesting. The authors have thoroughly explored the problems of integrating Architecture, Engineering and Construction (AEC) practices into Industry 6.0. The authors presented the development management concept within defining bifurcation levels. The paper has a sufficient level of novelty and practical recommendations. The methodology is adequate and clearly described. The results corresponded to the goal. The discussion is well conducted.
However, I would like to draw the authors' attention to the following. It’s better to use the full name at the first mention. This comment refers to Architecture, Engineering and Construction (AEC) in the title and in line 36. Moreover, I suggest that the abstract be shortened to 250 words so that potential readers can get a better idea of the information..
I wish you good luck!
Best regards,
The reviewer
Minor editing of the English language required
Author Response
Thank you for the constructive comments.
- The complete abbreviation "Architecture, Engineering, and Construction" (AEC) was added in the first paragraph.
- The abstract was reformed and reduced to 257 words.

Reviewer 3 Report
This paper explores the applications of industry 6.0 in the domain of architecture, engineering, and construction. The topic of this review paper is interesting. However, I'm confused about the literature review method used in this paper. The keywords are related to "Industry 4.0" and “Industry 5.0” in the literature review method. However, the title of this paper is associated with industry 6.0. It means that the topic of this paper might not match the literature review results. Moreover, the novelty of this paper needs to be clarified. More details about AEC industry 6.0 technologies should be provided. Detailed comments are listed as follows:
1. The authors should clarify the novelty of this paper through comparing this review paper with existing similar review papers. The following questions should be answered in this paper: What kinds of new perspectives do the authors offer in this paper? Why is this paper important for this domain?
2. The keywords of literature review are “construction and industry 4.0 and 5.0”, “Industry 4.0” “Industry 5.0,” “Society 5.0” and “Construction 5.0”. If the authors only utilize these keywords, the topic of this paper should be "construction and industry 4.0 and 5.0" rather than "Architecture, Engineering and Construction Industry 6.0". Why don’t the authors utilize the keywords associated with "Architecture, Engineering and Construction" and " Industry 6.0"?
3. The written English and organization of this paper should be improved. For instance, the name of section 5 is “Results and future AEC Industry 6.0 Research”. I think it is better to remove “results”, since this section doesn’t provide any results. Moreover, section 5.2 is about AEC Industry 5.0. This paper is associated with Industry 6.0. It might be better to remove this section if it is not associated with industry 6.0.
4. More details about industry 6.0 technologies should be provided. In this manuscript, the descriptions about industry 6.0 technologies are very shallow. The authors should introduce the principles, advantages and disadvantages of common 6.0 technologies in the AEC domain.
The written English and organization of this paper should be improved. For instance, the name of section 5 is “Results and future AEC Industry 6.0 Research”. I think it is better to remove “results”, since this section doesn’t provide any results. Moreover, section 5.2 is about AEC Industry 5.0. This paper is associated with Industry 6.0. It might be better to remove this section if it is not associated with industry 6.0.
Author Response
Thank you for the encouraging and constructive comments.
- The questions were answered at the end of the introduction.
- The keywords section was revised accordingly.
- The paper was proofread.
The section 5 title was changed to “5. Progressive Insights and Forward-Thinking Paradigms in AEC Industry 6.0 Research
It was an error in sub-section 5.2. The new title is “Human-Centric Evolution: Navigating the Transition to AEC Industry 6.0.”
- A new text section was implemented: Industry 6.0 technologies within the Architecture, Engineering, and Construction (AEC) sector have notable advantages. They provide extensive interconnectivity, optimizing worldwide partnerships. The integration of digital twins combines physical structures with real-time digital knowledge. Antifragile design methodologies have the potential to provide constructed environments that exhibit resilience and adaptability. The enhancement of building processes is facilitated by prioritizing software qualities such as openness and security. In general, this technological phenomenon enhances human knowledge and skills and the capabilities of construction equipment, creating opportunities for novel and environmentally conscious possibilities. On the other hand, the rise of Architecture, Engineering, and Construction (AEC) Industry 6.0 calls for significant investment in the economic, social, and technological infrastructures to ensure smooth integration. These technological developments can alter the nature of the workforce in the AEC industry, making certain positions obsolete and upending others. This change might exacerbate already-existing socioeconomic inequities and eliminate job possibilities, particularly for individuals who lack knowledge of this cutting-edge technology. Furthermore, the widespread use of AEC 6.0 can worsen environmental problems by causing re-source depletion and more pollution. Such consequences may threaten the sustainability of artificial ecosystems and the welfare of future generations. This study offers a crucial investigation of the revolutionary effects of Industry 6.0 on the Architecture, Engineering, and Construction (AEC) industry.

Round 2
Reviewer 3 Report
My comments have been responsed well.